# Decoding budget awareness: A multivariate analysis of generation Z undergraduates

**Gonca Güngör Göksu**[1¤a]*, **Erdal Eroğlu**[2], **Cihan Yüksel**[3], **Durdane Küçükaycan**[4¤b]

1 Department of Public Finance, Faculty of Political Sciences, Sakarya University, Sakarya, Türkiye,
2 Department of Public Finance, Biga Faculty of Economics and Administrative Sciences, Çanakkale Onsekiz Mart University, Çanakkale, Türkiye, 3 Department of Public Finance, Faculty of Economics and Administrative Sciences, Mersin University, Mersin, Türkiye, 4 Department of Public Finance, Faculty of Economics and Administrative Sciences, Eskişehir Osmangazi University, Eskişehir, Türkiye

¤a Current address: Department of Public Finance, Faculty of Political Sciences, Sakarya University, Sakarya, Türkiye
¤b Current address: Department of Public Finance, Faculty of Economics and Administrative Sciences, Eskişehir Osmangazi University, Eskişehir, Türkiye
* ggungor@sakarya.edu.tr

## Abstract

Understanding public finance and budget processes has become increasingly important for the younger generation in recent years. This knowledge enables Generation Z to participate more effectively in democratic life as informed citizens. However, there is limited research on Generation Z's awareness of public budgets in the existing literature. Therefore, the study addressed a significant gap in the public finance literature. We aimed to examine the determinants influencing the state budget awareness of Generation Z undergraduates by using the survey: "Citizens' Budget Awareness and Effective Factors". Data was collected from 3,972 undergraduates across all disciplines enrolled in four Turkish state universities between December 2023 and April 2024. We conducted a multiple linear regression analysis to examine the data. The main results showed that being informed about allocated public revenues for which public services, the constitutional power of the purse, budgeting processes and budget-related transactions were the determinants that most influenced the budget awareness of the students. On the contrary, being informed about the connection between taxes and public services, the Public Financial Management and Control Law No. 5018, and the amounts of public revenues within the state budget were determined as the lowest determinants influencing their awareness. Based on our results, we mainly suggested that (i) comprehensive education programmes and interactive learning should be organised in universities by using digital tools, (ii) the link between public revenues and expenditures should be explained through awareness campaigns, and (iii) budget processes should be made more transparent and accessible.

**Data availability statement:** The research has been funded within the TÜBİTAK (Scientific and Technological Research Council of Tuürkiye) Project (project no: 123K798), and the data that we obtained has been made available as open access on the project's official website. You can access the open data by clicking the link below and navigating to the document, "Data File". https://zvabbprojesi.sakarya.edu.tr/tr/icerik/24022/128056/ciktilar

**Funding:** This study has been supported by the Scientific and Technological Research Council of Türkiye (TUBİTAK) under project number 123K798 and called "Strengthening the Budget Awareness and Literacy Level of Generation Z: An Inter-University Approach" There was no additional external funding received for this study.

**Competing interests:** The authors have declared that no competing interests exist.

## Introduction

The budgeting process constitutes a fundamental instrument for governmental planning and decision-making and serves as a pivotal mechanism through which public funds are allocated, and societal resources are managed. As the cornerstone of fiscal management, budgeting enables governments to articulate their economic and political priorities, guiding the distribution of resources in alignment with broader policy objectives [1]. Beyond its technical function in economic and financial administration, the budget plays a crucial role in shaping social structures and influencing power dynamics. It reflects the relative distribution of economic and political power, reinforcing or challenging existing inequalities within society [2]. This regulatory capacity of the budget underscores the significance of citizens' direct participation in the budgeting process. Public involvement not only fosters transparency and accountability but also strengthens the democratic legitimacy of budgetary decisions [3,4]. Recognizing citizens' right to engage in and comprehend the budgeting process is essential for promoting inclusive governance. This concept, often referred to as the 'right to budget,' embodies a vital democratic principle that ensures individuals have a voice in the oversight and decision-making processes associated with public finance [5].

Reflecting this democratic ideal, the state budget is one of the most essential tools for governments to determine how to allocate resources collected on behalf of society, and the active participation of the public strengthens the democratic legitimacy of these processes. The participation of citizens in budget processes not only ensures that public resources are used more fairly and efficiently but also guarantees that budgeting is conducted in accordance with the principles of transparency and accountability [4–6]. Participatory budgeting practices enable citizens to oversee public spending, ensuring resources are allocated in alignment with community needs [7]. When transparency and accountability are realized, citizens may be less likely to evade or avoid paying their taxes, as these processes build trust and reinforce confidence in the proper use of public funds [8]. Research indicates that transparent budget processes enhance citizens' motivation to comply with tax obligations, thereby reducing tax evasion tendencies [9,10].

It is essential that individuals have an awareness of budgetary processes and possess the analytical capacity to understand and critically evaluate budget policies so that citizens can effectively participate in budgeting processes [11,12]. Active participation in budgeting processes can enhance individuals' public budget awareness [9]. In this regard, educational programs that enhance citizens' budget awareness and literacy have been observed to strengthen their engagement in fiscal processes. According to [7], the involvement of financially informed and educated citizens contributes to the democratic and transparent implementation of public policies by governments. Individuals with high economic awareness are better equipped to guide the implementation of policies that will yield long-term societal benefits, free from political short-sightedness. Budget awareness or literacy not only encompasses the understanding of the technical details of state budgets but also includes the ability of individuals to comprehend the economic and social consequences of fiscal policies. However, certain constraints, both from the administration and external bodies, hinder citizens from acquiring sufficient knowledge about budgetary issues. One of

the most significant non-administrative constraints is the level of individual effort. Specifically, lack of interest in the state budget coupled with limited budget awareness and literacy prevents citizens from recognizing their budgetary rights.

In line with this argument, budget literacy refers to the ability to read, decipher, and understand public budgets, allowing for effective citizen participation in the budget process [13], while budget awareness refers to citizens being engaged with and informed about state budget matters as outlined in the constitution and additional legal texts, possess an adequate understanding of public revenues and the public services funded by these revenues and diligently monitor developments related to the budget [14]. Although budget literacy and awareness are not wholly synonymous, they share two key features: (i) both involve knowledge of public budgets, including government spending, tax rates, and public debt as well as the ability to participate in the budget process with a practical understanding of everyday matters such as tax filings and accessing social benefits, and (ii) both require a basic comprehension of the economic, social, and political impacts of budget policies, along with the ability to identify stakeholders and understand when and how to contribute during the annual budget cycle [13]. All efforts to raise budget awareness and literacy in any society also contribute to achieving United Nations Sustainable Development Goal 16 – Promote Peaceful and Inclusive Societies for Sustainable Development. This goal aims to promote peaceful and inclusive societies, provide access to justice for all, and build effective, accountable and inclusive institutions at all levels, along with its targets, 16.6, 16.7, 16.8 and 16.10 [15].

Many studies explored the citizen participation and perceptions in budget processes through different concepts, such as budget rights, budget perception, budget literacy, and budget awareness [16,17]. Specifically, a significant number of previous studies found that the participants' perceptions of budgets are generally low [18–22]. Some studies found that budget literacy increases citizens' participation in budget processes [23–25]. However, there are limited studies examining the budget awareness and knowledge levels of Generation Z undergraduate students [16,22,26,27]. As future decision-makers and policymakers, their active participation in budgeting processes is crucial for democratic governance. Furthermore, growing up in the digital age, Generation Z is known to possess strong competencies in quickly accessing information through technology [28].

In this context, the involvement of youth in participatory and decision-making processes across various spheres of social life, particularly in matters that directly affect their emotions, thoughts, and behaviours, has become a significant research focus on the multiple disciplines in recent years [29–43]. In light of this growing interest, this study is thus motivated by the importance of enhancing budget awareness and literacy to increase Generation Z's ability to understand the social, economic, and political implications of budget policies.

The study aims to identify the determinants influencing the public budget awareness of Generation Z undergraduate students through multiple linear regression analysis. As a generation grown up in the digital era, Generation Z exhibits unique consumption patterns and behaviours; however, the existing literature offers only a limited number of studies examining this generation's budget awareness and participation in public fiscal processes. This study seeks to address a critical gap in the literature regarding Generation Z undergraduate students' budget awareness and participation in public fiscal processes and contribute to a deeper understanding of Generation Z's awareness levels regarding budget processes. In this context, the study's findings are intended to provide insightful data for policymakers and higher educational institutions, thereby fostering more active involvement of Generation Z undergraduate students in budgetary processes. The following section reviews the literature on budget awareness and literacy, which provides the foundation for the formulation of our hypotheses. Subsequently, we outline the methodology employed in the study and present the findings. In the concluding section, we analyse and discuss these results, propose policy recommendations, and provide a summary of the conclusions.

## Literature review and hypotheses

### Previous studies

The literature contains numerous studies exploring students' financial, economic, and tax literacy at both national and international levels [26,44–51] but there remains relatively limited research on state budget awareness and literacy. In

some studies, the concept of budget literacy has been limited to the ability to read and understand individual or corporate budgets [52–55]. However, budget awareness and literacy provide significant positive externalities, particularly in enhancing fiscal transparency, accountability, and public participation at both central and local levels [53]. Furthermore, a strong connection exists between participatory budgeting, poverty reduction, and improved access to essential services [56–58].

Participating in budgeting process can function as a "school of democracy", and it requires citizens to develop specific skills that help them enhance their budget awareness and literacy. Thus, informed citizens are better equipped to contribute meaningfully to the budgeting process and to make more effective decisions [25]. Despite the ongoing efforts to enhance fiscal information disclosure, our knowledge regarding how citizens understand and use public fiscal information remains limited. Hence, it is essential to conduct further research on budget awareness and literacy.

The existing studies have fundamentally discussed which measures could be taken to enhance citizens' budget awareness and literacy. With regard to more specific cases, [59] explored the use of information and communication technology to improve budget literacy among local residents, in Karanganyar Regency, Indonesia. Increasing the accessibility and development of information in Indonesia, consisting of thousands of islands geographically, is extremely important for increasing the existence and multiplication of the national asset [60]. Another study highlighted low public budget literacy as a barrier to fiscal transparency and accountability, proposing community-based education, document simplification, and digitalization to enhance citizen participation in Indonesia [61]. A group of researchers in South Korea developed a Budget Map application to enhance citizens' budget awareness country-wide [25]. In addition, the most comprehensive international study on budget literacy analysed 35 case studies from 34 countries [11]. They explored a range of activities aimed at improving budget literacy both within educational institutions and through extracurricular initiatives, offering valuable insights for other countries. The study regarding Russia explored how budget literacy could be increased at the regional level [62].

[14] calculated the budget awareness score of the citizens living in Sakarya province in Türkiye, revealing that citizens were not adequately interested in the budget topics or processes. [63], similarly, identified various factors affecting citizens' budget awareness levels in Türkiye. The studies focusing on South Korea searched the role of budget literacy in increasing citizens' participation in the budget process. They emphasised the positive relationship between accountability, participation, and budget literacy [64,65]. [66] examined the involvement of young citizens in the participatory budgeting process in the United States. [67] also highlighted that enhancing budget literacy and accountability is key to improving the efficiency of Kenya's health system by addressing resource mismanagement.

In terms of the relationship between education and budget awareness and literacy, focusing on how the Russian education process could be improved to increase the level of budget literacy of citizens, [68] outlined expectations for advancing the educational plans and programmes to enhance budget literacy among its citizens. [69], similarly, examined the development process of the "Citizens' Guide to the Federal Budget" in Russia, addressing the challenges of identifying target groups, simplifying budget information, and disseminating it to enhance budget literacy. [13] observed a Community Service Program in West Nusa Tenggara, Indonesia, teaching budget literacy and expenditure tracking to vulnerable groups using participatory and inquiry-based learning methods. [70] investigated the psychological and sociological factors undermining public budget awareness.

[71] examined how budget literacy and participation in universities enhance institutional performance through improved transparency, accountability, and resource management. [72] explored citizens' views on public finance education, finding a preference for starting in high school in Greece, and participants supported free online and on-site budget literacy seminars. To our best knowledge, no study has investigated the relationship between budget awareness and students in higher education or explored which determinants could impact their budget awareness levels yet. Therefore, this research aims to fill the gap in the literature by identifying the determinants that influence the state budget awareness among Generation Z undergraduate students.

**Hypotheses**

Building upon the literature, we determined the following hypotheses to achieve the study's aim:

H1: Being informed about the Citizen's Budget Guide and Citizen Final Account Report of Generation Z undergraduate students positively and significantly influences their budget awareness.

H2: Being informed about the state budget of Generation Z undergraduate students positively and significantly influences their budget awareness.

H3: Being enough interested in state budget issues of Generation Z undergraduate students positively and significantly influences their budget awareness.

H4: Being informed about the amounts and proportions of public revenues in the state budget of Generation Z undergraduate students positively and significantly influences their budget awareness.

H5: Being informed about the amounts and proportions of public expenditures of Generation Z undergraduate students positively and significantly influences their budget awareness.

H6: Being informed about allocating public revenues for which public expenditure items of Generation Z undergraduate students positively and significantly influences their budget awareness.

H7: Being informed about budgeting processes and budget-related transactions of Generation Z undergraduate students positively and significantly influences their budget awareness.

H8: Being informed about an official website that publicises the information and data of the state budget of Generation Z undergraduate students positively and significantly influences their budget awareness.

H9: Being informed about the Public Financial Management and Control Law No. 5018 of Generation Z undergraduate students positively and significantly influences their budget awareness.

H10: Being informed about the connection between taxes and public services of Generation Z undergraduate students positively and significantly influences their budget awareness.

H11: Being informed about the constitutional power of the purse of Generation Z undergraduate students positively and significantly influences their budget awareness.

H12: Being informed about allocating public revenues for which public services of Generation Z undergraduate students positively and significantly influences their budget awareness.

## Research methodology

### Data collection and sample selection

We collected data using the "Citizens' Budget Awareness and Effective Factors Survey," developed by [14]. This survey has been validated for reliability and validity in previous research. Before conducting our study, we submitted our proposal to the Social Science Ethics Committee at Sakarya University in 2023, and the relevant Ethical Committee approved the survey. This survey consisted of two main parts. While the first part included the determinants on budget awareness and calculated participants' budget awareness score, the second part focused on revealing the main factors. Fig 1 summaries the research process step by step.

 In our research, we administered the first part of the survey by employing undergraduates between December 2023 and May 2024 from four universities in Türkiye: Çanakkale Onsekiz Mart University (ÇOMU), Eskişehir Osmangazi University (ESOGU), Mersin University (MEU), and Sakarya University (SAU). We selected them because we could perform

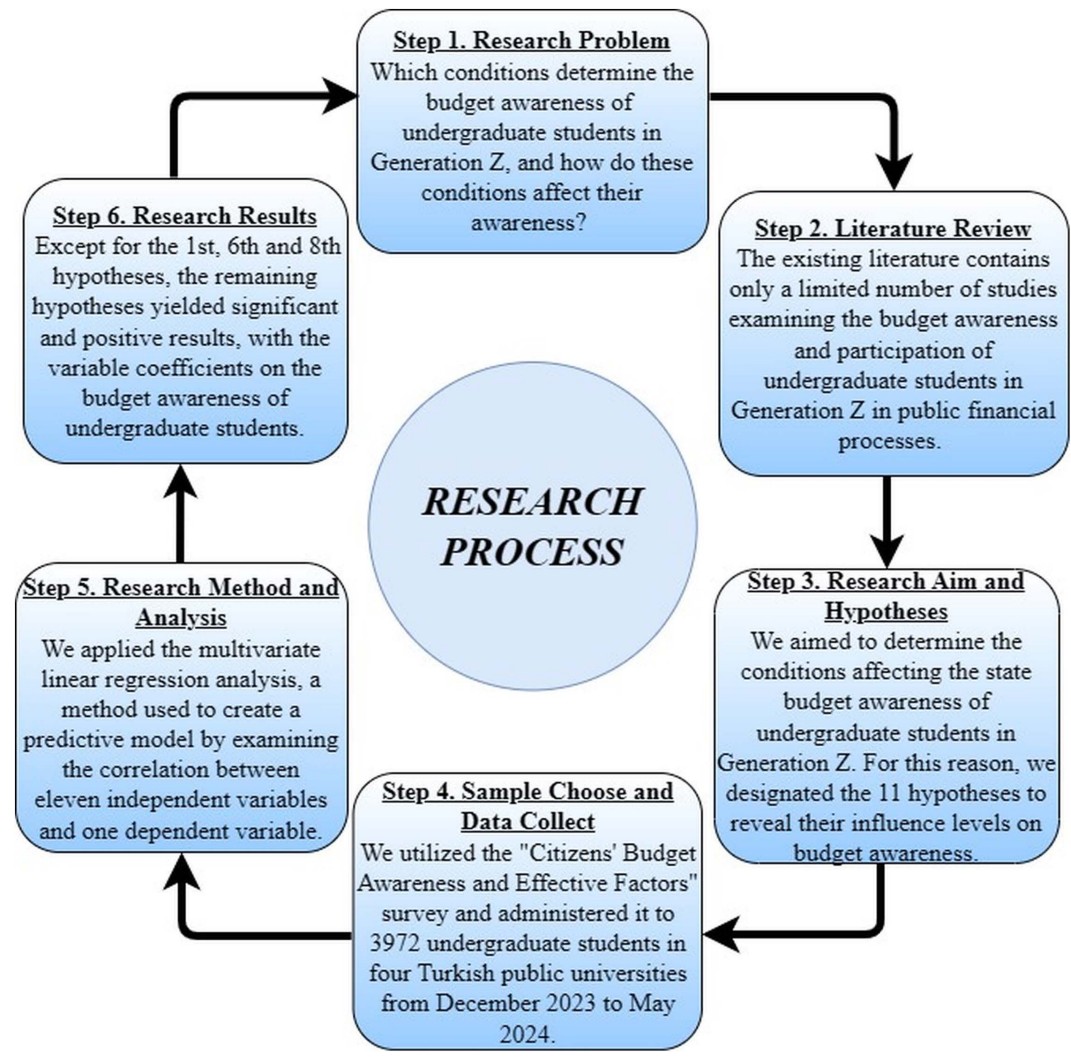

**Fig 1. Research process.**

the "Basic Budget Awareness" seminars in the 42 different education units in these universities. Each seminar was held for around 40 minutes, and our goal was to inform undergraduates about the state budget, the constitutional power of the purse, and other fundamental budget topics, e.g., public revenues and expenditures. We conducted the first part of the survey before the seminars and provided informed consent from participants at the introduction of the survey form. The participants voluntarily agreed to respond to the survey; thus, we could objectively measure the state budget awareness of the students since most of them had not taken any lessons, such as public finance or state budgets, during their education periods.

Other critical matters were calculating the sample size, determining the sample method and selecting which education units were included in the research. The total number of students in the four universities was 172.188, with 47.880 in ÇOMU, 30.100 in ESOGU, 40.909 in MEU, and 53.299 in SAU in 2023. In the research, the alpha level was 0.05, known as the 95% confidence limit, and the sampling margin of error was accepted as ±5. The target group included in the sample was determined by considering purposive sampling, one of the types of nonprobability sampling. Purposive sampling

is designed to align the sample more closely with the research aims and objectives, enhancing the study's rigour and the trustworthiness of the data and results. It is efficient in studies exploring specific knowledge domains [73]. This sampling technique have been preferred in other studies that searched the young individuals' opinions recently [74–77]. [78] also identified several sub-types of purposive sampling that researchers may consider, including snowball sampling, maximum variation sampling, extreme case sampling, and typical case sampling. Thus, maximum variation sampling, which we determined as sub-sampling in the research, focuses on identifying critical shared patterns that emerge from diverse situations, allowing researchers to derive significance from the heterogeneity present in the sample. The main reason for selecting this sub-sampling type was that the seminars were planned to be held in the units with at least 1000 undergraduates to reach the maximum number of participants. For this reason, we surveyed numerous undergraduates who have studied various scientific domains, such as social, science, engineering, health, education, art, and design, to synthesise diverse opinions and achieve more reliable findings and results. The sample size for each university was calculated at least 384 participants [79]. Additionally, we listed the education units according to the student numbers and chose the units with at least 1000 students at each university to reach as many students as possible. A total of 4171 students participated in the seminars and responded to the survey. Among them were 933 from ÇOMU, 1000 from ESOGU, 1109 from MEU and 1130 from SAU; however, 3972 responses were suitable for further analyses. Table 1 presents the details about demographic information, universities, and grades of participants as well as descriptive statistics.

## Measures and hypotheses

Each item in the scale with 13 items was scored on a five-point Likert scale from "Strongly disagree" to "Strongly agree". The reason for using a five-point Likert scale is ideal for gathering data from populations with educational systems using a 1–5 grading scale, as people are accustomed to evaluating within this range [80]. Budget awareness was determined as a dependent variable represented by the 13th item in the scale, "I consider that I have awareness regarding the state

**Table 1. Demographic profile of participants and descriptive statistics.**

|  | Frequency | Percentage | Median | Mode | Std. Variation |
|---|---|---|---|---|---|
| *University* |  |  |  |  |  |
| First University | 886 | 22,3 | 3,00 | 4 | 1,100 |
| Second University | 953 | 24,0 |  |  |  |
| Third University | 1057 | 26,6 |  |  |  |
| Fourth University | 1076 | 27,1 |  |  |  |
| *Degree* |  |  |  |  |  |
| 1st year | 1515 | 38,1 | 2,00 | 1 | 1,120 |
| 2nd year | 1064 | 26,8 |  |  |  |
| 3rd year | 810 | 20,4 |  |  |  |
| 4th year | 467 | 11,8 |  |  |  |
| More than four years | 91 | 2,3 |  |  |  |
| Missing Value | 25 | 0,6 |  |  |  |
| *Gender* |  |  |  |  |  |
| Men | 2455 | 61,8 | 1,00 | 1 | ,484 |
| Women | 1463 | 36,8 |  |  |  |
| Missing Value | 54 | 1,4 |  |  |  |
| *Total* | *3972* | *100* |  |  |  |

Source: Compiled by authors using the SPSS output.

budget." On the other hand, the first 12 items in the scale, presented in Table 2, represented the independent variables reflecting different determinants used effective in addressing the budget awareness of the participants.

The first item is related to the participants' awareness of the 'Citizen's Budget Guide', which aims to make the budget more understandable and evaluable for citizens. This guide has been publicized since 2017 by the Presidency of the Republic of Türkiye, Presidency of Strategy and Budget [81]. The Citizen Final Account Report has also been published since 2016 by the Republic of Türkiye Ministry of Treasury and Finance. This report presents the state's revenues and expenditures in a simple manner for citizens to understand [82]. In addition, the ninth item seeks to assess participants' awareness of Law No. 5018 on Public Financial Management and Control. This law encompasses significant regulations related to accountability, the cost-effective use of public resources, strategic planning, performance-based budgeting, multi-year budgeting, financial transparency, accrual-based accounting, and the reporting of accountability and financial statistics. It also covers internal control and both internal and external audits [83]. Table 3 presents the descriptive analysis results of independent and dependent variables.

Based on the mean values showed in Table 3, the participants' awareness for almost all items is below 3.00, except for the 10th and 12th items. This result indicates that their overall understanding of the budget is insufficient, with an average of 2.67 for the first 12 items (independent variables). The average of the 13th item (dependent variable) also supports this, at 2.79. Secondly, we assumed that the data are normally distributed because the Skewness and Kurtosis values for all items are between −1.5 and +1.5 [84]. In addition, the reliability value (Cronbach Alpha) of independent variables was calculated as 0,902 and raised to 0,912, including the dependent variable. The Cronbach Alpha Coefficients above 0.8 show excellent accuracy for the measures [85].

## Method and model development

We applied multiple linear regression analysis, a method used to create a predictive model by examining the correlation between two or more independent variables and one dependent variable to reach the study's aim. This method assumes a linear relationship between the independent and dependent variables [86,87]. Variables studied in real life are often affected by more than one variable, and multiple linear regression models are involved in this case [88]. The multiple linear regression line equation is generally written as $Y = a + bX + e$, where $Y$ is the dependent variable; $b$ is the slope or regression coefficient; is the intercept and $e$ is the error term as follows [89]:

Table 2. First 12 items used in model as dependent variables.

| Item Numbers | Items' Definition |
|---|---|
| 1 | I am informed about the Citizen's Budget Guide and Citizen Final Account Report. |
| 2 | I am enough informed about the state budget. |
| 3 | I am interested in the state budget issues. |
| 4 | I am informed about the amounts of public revenues in the state budget. |
| 5 | I am informed about the amounts of public expenditures in the state budget. |
| 6 | I am informed about allocating public revenues for which public expenditure items. |
| 97 | I am informed about budgeting processes and budget-related transactions. |
| 8 | I am informed about an official website that publicises the information and data of the state budget. |
| 9 | I am informed about the Public Financial Management and Control Law No. 5018. |
| 10 | I am informed about the connection between taxes and public services. |
| 11 | I am informed about the constitutional power of the purse. |
| 12 | I am informed about allocating public revenues for which public services. |

Source: Compiled by authors using the SPSS output.

**Table 3. Descriptive analysis of independent and dependent variables.**

| Variable | Mean | Std. Deviation | Variance | Skewness | Kurtosis |
|---|---|---|---|---|---|
| 1 | 2,31 | 1,073 | 1,152 | ,517 | -,405 |
| 2 | 2,65 | 1,064 | 1,132 | ,178 | -,635 |
| 3 | 2,98 | 1,211 | 1,468 | -,119 | -,952 |
| 4 | 2,50 | 1,070 | 1,145 | ,329 | -,602 |
| 5 | 2,50 | 1,080 | 1,166 | ,311 | -,681 |
| 6 | 2,74 | 1,138 | 1,295 | ,076 | -,908 |
| 7 | 2,58 | 1,080 | 1,166 | ,226 | -,710 |
| 8 | 2,62 | 1,260 | 1,587 | ,304 | −1,047 |
| 9 | 2,08 | 1,078 | 1,161 | ,948 | ,287 |
| 10 | 3,07 | 1,182 | 1,397 | -,262 | -,874 |
| 11 | 2,51 | 1,116 | 1,245 | ,370 | -,628 |
| 12 | 3,45 | 1,234 | 1,523 | -,537 | -,674 |
| 13 | 2,79 | 1,123 | 1,262 | ,096 | -,669 |

Source: Compiled by authors using the SPSS output.

$$Y_i = \beta_0 + \beta_1 X_{1i} + \beta_2 X_{2i} + \ldots + \beta_p X_{Pi} + \varepsilon i \ (i = 1, \ 2, \ \ldots, \ p) \tag{1}$$

Where Y represents the the dependent variable; $X_i$ denotes the independent variables; $b_0$ is the constant; $b_p$ are |regression coefficients estimated using the least squares method with n observations that measure the influence of the independent variables on the dependent variable ($i = 1, \ldots, p$). In Equation (1), ε represents a random variable that follows a normal distribution. It meets the fundamental assumption that its mathematical expectation (mean) is zero, and its variance is constant and independent of the variable X. The parameters β0, β1,..., βp are unknown, and it is assumed that p > 1 [88]. In multiple linear regression, we established a regression line between the independent and the dependent variables because we aimed to predict the budget awareness of Generation Z undergraduates using independent variables that reflect the determinants impact on the budget awareness increment. The model formula, which the abbreviation "D" refers as "Determinants" in accordance with the study's aim, is presented below:

$$\begin{aligned} \text{Bugdet Awareness} = & \beta_0 + \beta_1 D_1 + \beta_2 D_{2i} + \beta_3 D_3 + \beta_4 D_5 + \beta_4 D_4 + \beta_5 D_5 + \beta_6 D_6 \\ & + \beta_7 D_7 + \beta_8 D_8 + \beta_9 D_9 + \beta_{10} D_{10} + \beta_{11} D_{11} + \beta_{12} D_{12} + \varepsilon_i \end{aligned} \tag{2}$$

## Findings and discussion

### Findings

We collected the data through the "Citizens' Budget Awareness and Effective Factors" survey that included a total of 13 items. While the first 12 items, as detailed in Table 2, represented as independent variables, the last item was recognized the dependent variable. To analyse the data, we performed multiple linear regression analysis, a method well-suited for assessing the impact of independent variables on a dependent variable [86]. This analysis helps us to evaluate the influence of each independent variable on the dependent variable that represents the budget awareness of Generation Z. It also provided a comprehensive understanding of the relationships between the independent variables and the dependent variable [88]. The survey was responded to by 3972 undergraduates who enrolled in various education units at four universities. Before conducting the multiple linear regression analysis, we refined our data to eliminate outlier values and reduced the risk of potential outliers by considering Cooks' distance [90]. Cooks' distance, which should be less than 1, is

a diagnostic metric that evaluates how much the predicted values change when a single data point is removed, indicating that point's influence on the entire model [91]. Thus, 28 responses were removed from the dataset, and the multiple linear regression analysis was conducted with the remaining 3,944 responses. Among the respondents, 29.2% (1,152 undergraduates) were enrolled in the science and engineering sciences domain, 48.3% (1,905 undergraduates) in the social sciences domain, 11.8% (465 undergraduates) in the health sciences domain, 8% (316 undergraduates) in the educational sciences domain, and 2.7% (106 undergraduates) in the arts and architecture sciences domain. In addition, of the respondents, 38.4% (1,514 undergraduates) were in their first year, 27% (1,065 undergraduates) in their second year, 20.5% (809 undergraduates) in their third year, and 14.1% (556 undergraduates) in their fourth year. Following this step, we calculated the Pearson Correlation Analysis statistics, which provided preliminary insights into the relationships between the variables, as presented in Table 4.

According to Table 4, correlation analysis confirms a statistically significant relationship between the budget awareness (dependent variable) and all 12 determinants (independent variables). The study shows the moderate correlations between variables that are statistically significant at the 5% level. As long as the correlation values are below 0.80, there is no problem with multiple linearity [85]. Fig 2 illustrates the normal P-P of regression standardized residual and also residual plot for homoscedasticity assumption.

The verification of the model's assumptions is presented in Fig 2. The probability-probability plot (p-p plot) shows the values of the residuals with a linear pattern indicating the normality [92]. It also indicates homoscedasticity, which means the dependent variable has the same variance across all points of independent variables [89]. Table 5 gives a summary overview of the model.

In multiple linear regression analysis, one of the assumptions is that errors should be independent (or not autocorrelation problems) that refers using the Durbin-Watson test. A test result between 1,5 and 2,5 is acceptable [80]. The Durbin-Watson test value in the analysis is 1,947, indicating no detected autocorrelation problem. In the table, the R-value, 0,749, represents the correlation coefficient and highly indicates a correlation between budget awareness and effective determinants. The $R^2$ and adjusted $R^2$ values are calculated as 0,560 and 0,559, respectively. There is a slight difference between the $R^2$ and the adjusted $R^2$ values. Accordingly, the adjusted $R^2$ indicates the percentage of the change in the dependent variable explained by the independent variables [93]. 56% of the variation of the dependent variable is

**Table 4. Pearson correlation statistics.**

|  | 13 | 1 | 2 | 3 | 4 | 5 | 6 | 7 | 8 | 9 | 10 | 11 | 12 |
|---|---|---|---|---|---|---|---|---|---|---|---|---|---|
| 13 | 1,000 | ,431 | ,577 | ,457 | ,544 | ,556 | ,493 | ,551 | ,440 | ,427 | ,477 | ,561 | ,533 |
| 1 |  | 1,000 | ,550 | ,327 | ,511 | ,513 | ,354 | ,486 | ,391 | ,485 | ,275 | ,457 | ,207 |
| 2 |  |  | 1,000 | ,455 | ,605 | ,590 | ,499 | ,543 | ,403 | ,394 | ,416 | ,470 | ,360 |
| 3 |  |  |  | 1,000 | ,484 | ,436 | ,354 | ,388 | ,322 | ,259 | ,306 | ,310 | ,393 |
| 4 |  |  |  |  | 1,000 | ,744 | ,506 | ,548 | ,419 | ,471 | ,387 | ,483 | ,347 |
| 5 |  |  |  |  |  | 1,000 | ,564 | ,591 | ,426 | ,476 | ,411 | ,506 | ,370 |
| 6 |  |  |  |  |  |  | 1,000 | ,555 | ,374 | ,350 | ,506 | ,430 | ,384 |
| 7 |  |  |  |  |  |  |  | 1,000 | ,460 | ,473 | ,412 | ,532 | ,337 |
| 8 |  |  |  |  |  |  |  |  | 1,000 | ,469 | ,408 | ,467 | ,310 |
| 9 |  |  |  |  |  |  |  |  |  | 1,000 | ,318 | ,555 | ,183 |
| 10 |  |  |  |  |  |  |  |  |  |  |  | ,510 | ,462 |
| 11 |  |  |  |  |  |  |  |  |  |  |  | 1,000 | ,379 |
| 12 |  |  |  |  |  |  |  |  |  |  |  |  | 1,000 |

Note: Correlation is significant at the 0.05 level.

Source: Compiled by authors using the SPSS output.

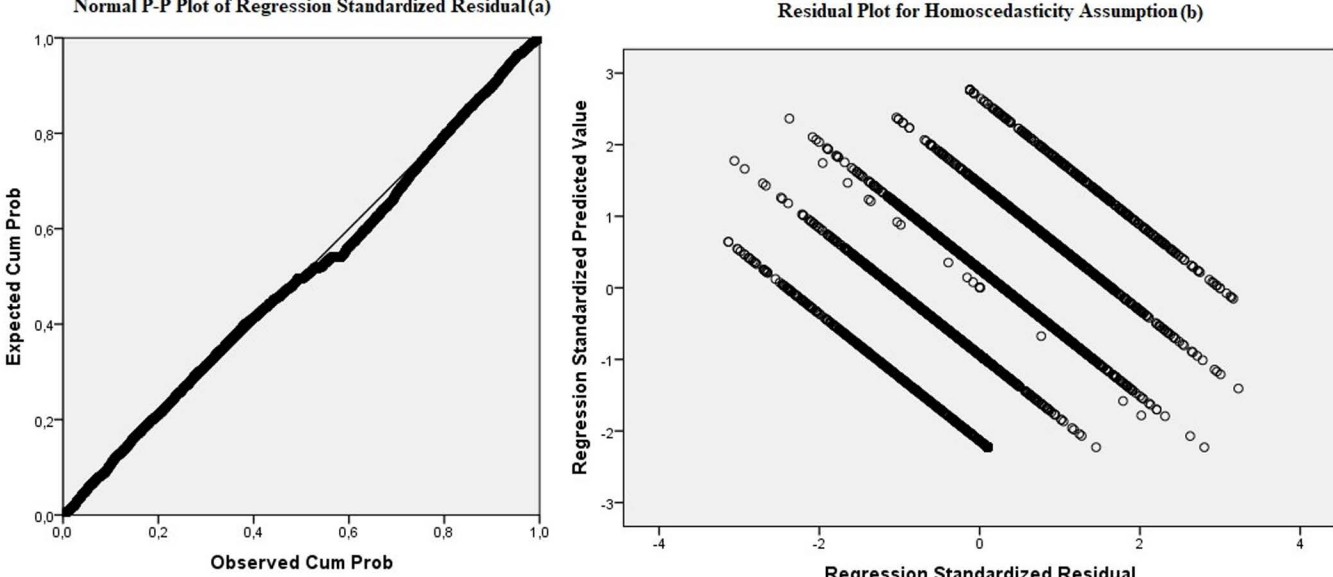

**Fig 2. Verification of model assumptions.** (a) normality of residuals, (b) independence of residuals.

**Table 5. Model summary.**

| Model | R | $R^2$ | Adjusted $R^2$ | Std. Error of the Estimate | Durbin-Watson |
|---|---|---|---|---|---|
| 1 | ,749[a] | ,560 | ,559 | ,740 | 1,947 |

a. Predictors: (Constant), D1, D2, D3, D4, D5, D6, D7, D8, D9, D10, D11 and D12.

Source: Compiled by authors using the SPSS output.

accounted for by the independent variables. In addition, the regression results show that the adjusted $R^2$ value is close to 0.60, indicating that the model fits enough well with the sample observations [88]. Table 6 illustrates the ANOVA (analysis of variance) statistics, including F and significant values of the research model.

In the ANOVA Table, the value of Fisher (F) test explains whether the regression model fits well with the data [94]. Additionally, the statistical significance of the regression model, less than 0,005, gives an overall signal that the model has a considerable capacity to statistically forecast the outcome variable, as shown by the analysis of variance (ANOVA). In Table 6, the value of the F test is 417,969 and p<,005. Thus, the regression model fits well with the data and at least one independent variable is significantly related to budget awareness. Table 7 presents the Beta coefficients (β) and also VIF values in the model.

**Table 6. ANOVA statistics.**

| Model | | Sum of Squares | df | Mean Square | F | Sig. |
|---|---|---|---|---|---|---|
| 1 | Regression | 2746,612 | 12 | 228,884 | 417,696 | ,000[b] |
| | Residual | 2154,066 | 3931 | ,548 | | |
| | Total | 4900,678 | 3943 | | | |

b. Predictors: (Constant), D1, D2, D3, D4, D5, D6, D7, D8, D9, D10, D11 and D12.

Source: Compiled by authors using the SPSS output.

**Table 7. Beta coefficients for the budget awareness and VIF values.**

| Model | Unstandardized Coefficients | | Standardized Coefficients | t | Sig. | Collinearity Statistics | |
|---|---|---|---|---|---|---|---|
| | β | Std. Error | | | | Tolerance | VIF |
| Constant | -,117 | ,045 | | −2,630 | **,000** | | |
| D1 | ,012 | ,015 | ,011 | ,801 | **,423** | ,566 | 1,766 |
| D2 | ,181 | ,016 | ,173 | 11,267 | **,000** | ,476 | 2,099 |
| D3 | ,078 | ,012 | ,084 | 6,555 | **,000** | ,675 | 1,482 |
| D4 | ,056 | ,018 | ,053 | 3,080 | **,002** | ,373 | 2,682 |
| D5 | ,062 | ,018 | ,060 | 3,374 | **,001** | ,358 | 2,791 |
| D6 | ,035 | ,014 | ,036 | 2,476 | **,013** | ,533 | 1,876 |
| D7 | ,113 | ,016 | ,109 | 7,210 | **,000** | ,485 | 2,060 |
| D8 | ,032 | ,012 | ,036 | 2,720 | **,007** | ,640 | 1,562 |
| D9 | ,051 | ,015 | ,049 | 3,495 | **,000** | ,560 | 1,786 |
| D10 | ,041 | ,013 | ,044 | 3,128 | **,002** | ,578 | 1,731 |
| D11 | ,161 | ,015 | ,160 | 10,657 | **,000** | ,494 | 2,024 |
| D12 | ,220 | ,012 | ,243 | 18,932 | **,000** | ,677 | 1,477 |

Source: Compiled by authors using the SPSS output.

VIF value is used to discover multicollinearity, and they should ideally be less than 4 (VIF < 4), which means that multicollinearity is not a problem in the model [95]. Based on the VIF values (ranging from 1,477–2,791) presented in Table 7, multicollinearity has not been detected in the model. The β coefficient provides estimates for the intercept and slopes of predictor variables [96]. As seen in Table 7, the β coefficients vary according to the variables in the model. The constant term in the model is negative and significant (-,117 and p < ,005). Except for D1, D6, and D8, the remaining Ds are significant and positive on determining budget awareness of the undergraduates because of p < ,005.

## Discussion

For Hypothesis 1, 6, and 8, the empirical findings reveal no significant and positive correlation between independent and dependent variables. More deeply, Determinant 1, being informed about the Citizen's Budget Guide and Citizen Final Account Report of the undergraduates, has not positively and significantly impacted their budget awareness. The possible explanation could be that Generation Z undergraduates did not adequately understand the documents called the Citizen's Budget Guide and the Citizen Final Account Report. Being informed about allocating public revenues for which public expenditure items and also being informed about an official website that publicises the information and data of the state budget have not been accepted as influence determinants on their budget awareness. Having budget information may not always influence citizens' opinions or expectations positively. Similarly, [97] found a negative relationship between budget information and budget satisfaction in China. In addition, government budget information disseminated through various forms of online mass media, such as web portals, can adversely affect citizens' perceptions of their government and budget processes [98]. On the contrary, there have been significant positive correlations between the budget awareness and the remaining determinants in the scale; therefore, we cannot reject these hypotheses: H2, H3, H4, H5, H7, H9, H10, H11 and H12. Based on Table 7, the regression model has been formulated using the β coefficients as follows:

$$\text{Budget Awareness} = -,117 + ,181D2 + ,078D3 + ,056D4 + ,062D5 + ,113D7 + ,051D9 + ,041D10 + ,161D11 + ,220D12$$

(3)

According to the equation, the top-ranked decision criteria (Ds) are D12, D2, D11, and D7, with coefficients of 0.220, 0.181, 0.161, and 0.113, respectively. Notably, D12 has the most substantial positive influence on budget awareness.

 

Regarding formula three, the top-ranked decision criteria (Ds) are D12, D2, D11, and D7, with coefficients of 0.220, 0.181, 0.161, and 0.113, respectively. Notably, D12 has the most substantial positive influence on budget awareness. Each one-unit increase in informing about the allocation of public revenues to public services results in a 22.0% increase in budget awareness.

For every one-unit increase in D2, which represents being informed about the state budget, the awareness level increases by 18.1%. These findings demonstrate significant positive influences of state budget information on budget awareness and are in line with the literature. Because both budget allocations improve public services' performance [99], and also citizens' budget information about allocations positively affects public service performance through increased citizen trust [97]. Furthermore, public governance effectiveness, directly linked to the performance of the public sector, the quality and quantity of public services, and the efficient allocation of public resources, plays a crucial role in achieving citizen satisfaction and awareness [100].

D11, representing the awareness of the constitutional power of the purse, shows a strong positive effect, with each one-unit increment in this item leading to a 16.1% rise in budget awareness. This finding indicates that Generation Z undergraduates recognize that the authority over budgetary decisions, commonly called the "power of the purse," resides with legislature representing them. The budget is the key law for any legislature, and citizens are aware the results of by the disruptions to public services caused by a delayed budget. For this reason, they are willing for the legislative to pass the budget on time [101]. [22] found that undergraduates' perceptions of the power of the purse were influenced by education, transparency, efficiency in public financial management and perceptions of public expenditures. Therefore, as citizens' representatives, parliaments should ensure that the budget revenue and expenditure measures are fiscally sound and implemented efficiently and correctly [102].

Moreover, for every incremental increase in D7, which signifies the importance of being well-informed about budgeting processes and transactions related to the budget, there is a corresponding 11.3% rise in budget awareness among under-graduates. This finding directly addresses the significance of participation in budget processes. [103] ascertained that the one determinant associated with increasing citizen participation in public affairs is high educational level. Citizens' under-standing of budget processes is linked to increased involvement in budget-related transactions, increasing public involve-ment in the budgetary processes can build trust between the government and the public by giving individuals a voice and enabling them to influence decisions [104].

In the multi-regression analysis, the determinants with the most minor influence on budget awareness among the participants were D10, D9, and D4, with coefficients of 0.041, 0.051, and 0.056, respectively. D10 measured the impact of informing undergraduates about the connection between taxes and public services, revealing a modest increase in budget awareness of only 4.1% for each unit increase. Young individuals without any taxation course experience presented lower tax literacy and morale than their contemporaries who received tax education [105], and this result proved that tax edu-cation improves their motivations regarding public finance and taxes [106]. In our research, most participants had never studied taxes during their education period, too. Tax revenues provide funding that benefits the common good [107]. Enhancing undergraduates' tax awareness is essential because higher tax knowledge also correlates positively with stu-dents' attitudes towards taxes and increased participation in public budget processes [108]. Similarly, [109] detected that higher education programs could be evaluated for their alignment with labour market needs, highlighting the importance of incorporating tax education into undergraduate curricula.

D9, which pertains to awareness of Public Financial Management and Control Law No. 5018, regarded as the fiscal constitution that aligned Türkiye's public financial management system with international standards [110], has had a slight effect, enhancing budget awareness by just 5.1%. The financial regulations enshrined in the constitution and other statutory laws play a pivotal role in shaping the state's fiscal standing and autonomy, impacting elements such as budget-ary processes, public debt, and state-owned enterprises [111]. Therefore, laws and regulations should guarantee equal access to administrative processes for all citizens [112].

D4, which concerns knowledge about the total amount of public revenues in the state budget, has shown a minimal influence, with a 5.6% increase in awareness per unit. Understanding the totality of public revenues is crucial for effective budget planning and transparency and allows governments to allocate resources efficiently and meet public needs [113].

D3 and D5, which involved interest in state budget issues and knowledge about the amounts of public expenditures, respectively, have presented moderately more robust impacts at 7.8% and 6.2%, underscoring a relative but discernible interest in broader budgetary details compared to the lower-ranked Ds. These findings illustrate that the visibility of budget revenues and expenditures is limited from a young citizens' perspective, necessitating practical measures to enhance budget transparency [114]. Specifically, youth participation is a key way for them to express their interests and impact society [115]. Informing and inviting young citizens to the budget processes will enhance their awareness of the impact of their public revenues, primarily taxes, on the distribution of public expenditures [116]. Furthermore, citizens can increase trust in government financial processes when they are aware that public services are managed with transparency [112]. Public involvement in budget management can significantly raise trust in public authorities, enhancing the efficiency and effectiveness of budget revenues and expenditures [117]. As a result, citizen participation in public affairs provides a better assessment of their preferences regarding the impacts of government policies, which can inform governments in aligning their decisions with citizens' interests [17].

## Conclusion and recommendations

This research aimed to understand the key determinants influencing budget awareness of Generation Z undergraduate students by employing multiple linear regression analysis. Recognizing the crucial role of budget literacy and awareness in fostering citizen participation, this study focused on identifying the factors that enhance or impede the awareness of Generation Z undergraduates. First of all, the study contributed to the existing literature on budget awareness, literacy and public participation by identifying the key determinants influencing budget awareness among Generation Z undergraduate students. It provided a detailed understanding of how specific areas of knowledge, such as allocating public revenues to services, the complexities of state budget processes, and the constitutional mechanisms of fiscal control, affect young adults' budget awareness. Considering the limited focus on this demographic in current budget literacy research, this exploration has been significant. By emphasized how Generation Z's digital fluency and unique socio-economic perspectives intersected with fiscal awareness, the research offered a theoretical framework that could be used to assess and improve educational strategies in public financial management. This framework highlighted the direct impact of specific knowledge on budget awareness and illustrated the broader socio-political engagement it encourages among Generation Z citizens.

This study not only identifies the determinants of budget awareness but also offers recommendations for the development of these determinant criteria, highlighting the potential externalities that these developments may create. Therefore, the study has three significant widespread implications: (a) providing policymakers with a roadmap to strengthen governance channels, (b) assisting educational institutions in creating a curriculum that contributes to budget awareness, and (c) offering ideas for behavioural economics studies in this field, particularly those using nudging and framing techniques, by providing insights based on the findings of this study.

The primary findings indicated that specific determinants significantly enhanced budget awareness among Generation Z undergraduate students. These determinants included being well-informed about how public revenues are allocated to public services (D12 decision criterion, with a coefficient of 0.220 and a significance level of 0.000), possessing knowledge of the state budget (D2 decision criterion, with a coefficient of 0.181 and a significance level of 0.000), understanding the constitutional power of the purse (D11 decision criterion, with a coefficient of 0.161 and a significance level of 0.000), and having knowledge about budgeting processes and budget-related transactions (D7 decision criterion, with a coefficient of 0.113 and a significance level of 0.000). These elements collectively contributed to a substantial positive impact on students' budget literacy, particularly their ability to discern how governmental fiscal policies influence public welfare

and resource distribution. This situation heightened awareness among students reflects a robust grasp of fiscal responsibility and governance, underscoring the importance of integrating these topics into educational curricula to cultivate a well-informed, responsible future electorate. Accordingly, the results suggested that the students' awareness of how public revenues are used for services such as education, healthcare, and environmental protection performs a pivotal role in shaping their understanding of the budgeting process. The connection between public revenues and public services resonated strongly with students, indicating that they were more likely to engage with fiscal information when it directly relates to the services they interact with in their daily lives. It supported the previous literature [23,25] on the role of budget literacy in promoting active citizen engagement, especially among younger generations who are more aware of social and economic impacts.

The moderate determinants of budget awareness among Generation Z undergraduate students are interest in government budget issues (D3 decision criterion, with a coefficient of 0.078 and a significance level of 0.000) and knowledge of the amount of public expenditures (D5 decision criterion, with a coefficient of 0.062 and a significance level of 0.001). These criteria may contribute to students' development of a deeper understanding of budget management. These findings indicate that educational strategies aimed at enhancing budget awareness should focus on both providing knowledge about the government budget and fostering awareness of public expenditures.

The findings also identified several determinants that had relatively low impact on enhancing budget awareness among the participants. It included a basic understanding of the connection between taxes and public services (D10 decision criterion, with a coefficient of 0.041 and a significance level of 0.002), which suggests that merely knowing about tax allocations does not significantly influence students' awareness of budgetary matters. Similarly, knowledge with the Public Financial Management and Control Law No. 5018 (D9 decision criterion, with a coefficient of 0.051 and a significance level of 0.000), which serves as a cornerstone of fiscal governance in Türkiye by aligning the national financial management system with international standards, showed limited effect on students' budget consciousness. Additionally, knowledge regarding the overall amounts of public revenues within the state budget (D4 decision criterion, with a coefficient of 0.056 and a significance level of 0.002) was also found to be a less influential determinant in increasing budget awareness, indicating that broader financial figures without specific contextual information may not substantially impact students' understanding of fiscal policies.

The findings elucidate the Generation Z undergraduate students' marked preference for allocating public revenues towards public services rather than mere expenditure items, underscoring their awareness of the link between government spending and its societal impact. This insight into functional classification, a method that organizes government activities by purpose, reveals that these students favour spending that directly enhances community services, reflecting a mature understanding of fiscal policy and its implications for public welfare. Furthermore, their heightened budget awareness is bolstered by a comprehensive grasp of budgetary governance, including the state budget processes, constitutional power of the purse, and budget-related transactions. Based on these results, the link between public revenues and expenditures should be explained through awareness campaigns. Comprehensive education programmes and interactive learning should also be organised in universities by using digital tools. The findings suggest that educational programs focusing on fiscal policy and public spending can significantly enhance civic engagement and advocacy for effective and transparent governance to promote a more accountable government by allowing citizens to assess government activity's actual costs and benefits and understand public performance. This situation makes a strong case for curricula integrating these components to foster a well-informed electorate among emerging adults.

The findings also provided practical insights for policymakers and educational institutions looking to enhance public financial management and budget education. The study explained the importance of tailoring educational content to include detailed discussions on public budgeting processes, constitutional rights related to budgeting, and the societal impacts of fiscal decisions. These insights have been especially valuable for creating curricula that boost budget literacy but also effectively engaged Generation Z students, capitalizing on their preference for digital platforms and transparency.

Additionally, the identification of less influential factors pointed to areas where educational interventions might need to be improved or restructured. The study ensured that fiscal education has been both thorough and engaging. Such insights can help guide the development of targeted educational programs and public awareness campaigns that address the gaps in understanding and engagement observed among students.

It might present that while students understand the broad concepts of public finance, they may lack a deeper comprehension of the more complex legal and procedural aspects of budgeting. Bridging this gap is crucial, as understanding budget laws and processes is essential for fostering a well-rounded fiscal awareness that supports meaningful participation in public financial decision-making. The study also highlighted the broader societal and educational implications of budget awareness among Generation Z. As digital natives, these students are highly skilled at accessing information quickly, yet their ability to critically engage with and apply this information in real-world contexts, such as public finance, is still developing. Therefore, educational institutions need to incorporate budget awareness and literacy programs that inform students about the technical aspects of public budgeting and emphasise the importance of democratic participation and accountability in financial management. In addition, students equipped with the knowledge to understand and analyse budgetary decisions are more likely to hold governments accountable for using public funds. Furthermore, by fostering an understanding of the connection between fiscal policies and social outcomes, they can contribute to formulating policies that promote fairness, efficiency, and sustainability in allocating public resources.

The findings provided valuable insights for policymakers and educators aiming to enhance budget awareness among young citizens. The research underscores the need for targeted interventions that not only provide access to fiscal information but also cultivate the skills required to interpret and apply that information effectively. By doing so, it is possible to cultivate a generation of informed and engaged citizens who can play an active role in the budgeting process and contribute to the broader goals of transparency, accountability, and democratic governance. Another significant recommendation concerns the disconnect between technical budget information availability and students' interest or ability to engage with these resources. While technological tools like official government websites are intended to promote transparency, the results suggest that more than simply providing information is required. Therefore, budget processes should be made more transparent and accessible, and students should be guided and educated on interpreting and using this information effectively.

In conclusion, increasing the level of budget awareness and literacy among Generation Z will be an essential step towards ensuring social justice and economic participation. Raising individuals who comprehend the social consequences of budget policies and can critically approach fiscal processes will contribute to economic welfare and the strengthening of democratic participation. Improving the budget awareness and literacy of Generation Z should also be considered a strategic goal for social sustainability in connection with the United Nations Sustainable Development Goals 16 – Promote Peaceful and Inclusive Societies for Sustainable Development.

## Acknowledgments

We would like to express our gratitude to the administrations of Çanakkale Onsekiz Mart University, Eskişehir Osmangazi University, Mersin University, and Sakarya University for granting us permission to conduct the survey. We also extend our thanks to the undergraduate students enrolled in these universities for their valuable contributions to our research through their opinions.

## Author contributions

**Conceptualization:** Gonca Güngör Göksu, Erdal Eroğlu, Cihan Yüksel, Durdane Küçükaycan.

**Data curation:** Gonca Güngör Göksu, Erdal Eroğlu, Cihan Yüksel, Durdane Küçükaycan.

**Funding acquisition:** Gonca Güngör Göksu.

**Investigation:** Gonca Güngör Göksu, Erdal Eroğlu, Cihan Yüksel, Durdane Küçükaycan.

**Methodology:** Gonca Güngör Göksu.

**Writing – original draft:** Gonca Güngör Göksu, Erdal Eroğlu, Cihan Yüksel, Durdane Küçükaycan.

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
