## [Decision Letter · Decision Letter 0]

2 Mar 2025

Dear Dr. Göksu,

Thank you for submitting your manuscript to PLOS ONE. After careful consideration, we feel that it has merit but does not fully meet PLOS ONE’s publication criteria as it currently stands. Therefore, we invite you to submit a revised version of the manuscript that addresses the points raised during the review process.

Generally, the majority of the reviewers thought your manuscript has some merits. However, certain issues require your careful attention. Please try address the reviewers' comments as much as you can. I wish you all the best.

We look forward to receiving your revised manuscript.

Kind regards,

Tatchalerm Sudhipongpracha

Academic Editor

PLOS ONE

Journal Requirements:

“This study has been supported by the Scientific and Technological Research Council

of Türkiye (TUBİTAK) under project number 123K798 and called “Strengthening the

Budget Awareness and Literacy Level of Generation Z: An Inter-University Approach””

“This study has been supported by the Scientific and Technological Research Council of Türkiye (TUBİTAK) under project number 123K798 and called “Strengthening the Budget Awareness and Literacy Level of Generation Z: An Inter-University Approach”.”

“This study has been supported by the Scientific and Technological Research Council

of Türkiye (TUBİTAK) under project number 123K798 and called “Strengthening the

Budget Awareness and Literacy Level of Generation Z: An Inter-University Approach””

Reviewers' comments:

Reviewer's Responses to Questions

**Comments to the Author**

1. Is the manuscript technically sound, and do the data support the conclusions?

Reviewer #1: Yes

Reviewer #2: Yes

Reviewer #3: No

2. Has the statistical analysis been performed appropriately and rigorously?

Reviewer #1: Yes

Reviewer #2: I Don't Know

Reviewer #3: No

3. Have the authors made all data underlying the findings in their manuscript fully available?

Reviewer #1: Yes

Reviewer #2: Yes

Reviewer #3: No

4. Is the manuscript presented in an intelligible fashion and written in standard English?

Reviewer #1: Yes

Reviewer #2: No

Reviewer #3: No

Reviewer #1: This article is excellence, the structure and point of discussion is on point. This article just need to put some theory in discussion for enrich the discussion and result part. I don't see any major revision for this article and also this article relate to the scoop of this journal.

Reviewer #2: Here are some suggestions for improving your paper after reading it. These brief remarks should serve as motivation for you to promptly enhance the work that is being submitted for publication in the PLOS ONE Journal. I hope it is always successful and helpful.

Reviewer #3: Comments:

Firstly the authors used simple age old multiple linear regression, which is not a mutivariate analysis; there is nothing multivariate in the whole study. The heading of the paper is misleading. The method is outdated in this era of machine learning.

secondly, the authors used purposive sampling with determination of sample sizes, which seems like an outdated method to deal with real-life participants with so many random and model-based techniques available for sampling.

**Do you want your identity to be public for this peer review?** For information about this choice, including consent withdrawal, please see our Privacy Policy

Reviewer #1: **Yes: **

Reviewer #2: **Yes: ** Buyamin

Reviewer #3: **Yes: ** Kaustav Aditya, Senior Scientist, ICAR-IASRI, New Delhi

---

## [Author Response · Author response to Decision Letter 1]

14 Mar 2025

Decoding Budget Awareness: A Multivariate Analysis of Generation Z Undergraduates

Respected Reviewers,

We would like to thank you for your careful review of our paper and for the comments, corrections, and suggestions that helped us to improve this study. We have carried out revisions in line with the recommendations/suggestions. By doing so, we believe that the paper has been significantly improved. We have coloured our edits red in the text. Please read our responses below:

GENERAL COMMENTS AND RESPONSES

Reviewer #1: This article is excellence, the structure and point of discussion is on point. This article just need to put some theory in discussion for enrich the discussion and result part. I don't see any major revision for this article and also this article relate to the scoop of this journal.

Response 1: Dear Reviewer 1, we sincerely thank you for your valuable and supportive feedback. Your encouraging opinions are greatly appreciated for us and motivate us to further improve ours study.

Reviewer #2: Here are some suggestions for improving your paper after reading it. These brief remarks should serve as motivation for you to promptly enhance the work that is being submitted for publication in the PLOS ONE Journal. I hope it is always successful and helpful.

Response 2: Dear Reviewer 2, we sincerely thank you for your valuable and supportive feedback. We have revised the manuscript in line with your suggestions, and the changes have been highlighted in the text for your convenience.

Reviewer #3: Comments:

Firstly the authors used simple age old multiple linear regression, which is not a multivariate analysis; there is nothing multivariate in the whole study. The heading of the paper is misleading.

The method is outdated in this era of machine learning.

Secondly, the authors used purposive sampling with determination of sample sizes, which seems like an outdated method to deal with real-life participants with so many random and model-based techniques available for sampling.

Response 3: Dear Reviewer 3, we sincerely thank you for your constructive suggestions. We have revised the title and text accordingly. First, we realized that we had used an incorrect technical term. Therefore, we have replaced 'multivariate regression' with 'multiple regression' to ensure accuracy. In addition, you criticized we used an outdated method. In the literature, we examined the recent articles with multiple regression methods and we also cited them, which were indexed in the WoS database and published recently. For example;

• Bendo, A. and Brovina, F. (2024). “A statistical model using multiple regression analysis to predict equilibrium and sway index”. Journal of Physical Education and Sport, 24(6), 1446–1456.

• Singh, M., Ghai, S., Mishra, A.K., and Goyal, N. (2024). “Analysis of EAC using multiple regression and conditional process: A statistical approach”. Journal of Reliability and Statistical Studies, 17(1), 109–136.

• Wu, Q. and Qiu, Y. (2019). “Research on the Influencing Factors of China’s Logistics Industry Based o and Multiple Regression Model”. 2019 International Conference on Economic Management and Model Engineering (ICEMME), 360–363.

• Xue, W. and Sun, S. (2019). “Relationship between organizational improvisation and organizational creativity under multiple regression analysis”. Revista de Cercetare Si Interventie Sociala, 65, 206–229.

The sampling choice depended on the aim, sources, and context of the research. We aimed to reach the maximum number of Generation Z undergraduates in our study. For this reason, as mentioned in the text, we conducted the "Basic Budget Seminars" in the most crowded education units with at least 1000 students. Purposive sampling is designed to align the sample more closely with the research aims and objectives, enhancing the study's rigour and the trustworthiness of the data and results. This approach is efficient in studies exploring specific knowledge domains (Denieffe, 2020; Tonga, 2007). This sampling technique have been preferred in other studies that searched the young individuals’ opinions recently (Junus et al., 2023; Raut and Kumar, 2023; Irfan et al., 2024; Lin et al, 2024).

Palinkas et al. (2015) identified several sub-types of purposive sampling that researchers may consider, including snowball sampling, maximum variation sampling, extreme case sampling, and typical case sampling. Accordingly, maximum variation sampling, which we determined as sub-sampling in the research, focuses on identifying critical shared patterns that emerge from diverse situations, allowing researchers to derive significance from the heterogeneity present in the sample. For this reason, we surveyed numerous undergraduates who have studied various scientific domains, such as social, science, engineering, health, education, art, and design, to synthesise diverse opinions and achieve more reliable findings and results.

OTHER COMMENTS FROM REVIEWERS

1. I express my highest appreciation to the author, who has made many improvements by incorporating various inputs from previous reviewers.

Response: Dear Reviewer, we sincerely thank you for your valuable and supportive feedback. We have carefully paid attention all your suggestions.

2. Article structure:

The Title, Abstract, Introduction, Literature Review, and Research Methodology groups consist of 3,594 words (43.58%) of the total paper. The Results and Discussion, Conclusion, and Recommendations groups consist of 2,749 words (33.33%) of the total paper. References consist of 1,904 words (23.09%) of the total paper. In my opinion, the composition of this paper is not proportional because the Core Section of the Paper (Results, Discussion, and Conclusion) is less than 50.00% of the total paper.

Response: Dear Reviewer, we have enhanced the manuscript, especially the findings, discussion, and conclusion, according to all your suggestions. The titles “Findings and Discussion & Conclusion and Recommendations” have increased to 3.800. We have also supported the findings and discussion section with recently references.

3. Originality

This study has a unique theme with the state budgeting theme for Generation Z undergraduate students in four Turkish state universities. The study's results may be a reference for other countries for budgeting for students in government universities.

Response: We sincerely thank for your supportive opinion. We believe that this study will be a valuable reference for other researchers who are interested in public budget awareness and literacy.

4. Abstract

The author has compiled an abstract containing the objectives, methods, findings, implications, contributions, and recommendations accompanied by keywords representing the paper's discussions. However, the author has not included the background of this research.

Response: Dear Reviewer, we have included the background of this research in order to emphasize why we conducted this study and why understanding and having enough knowledge about public budgets by Generation Z is essential. In addition, we added the main suggestions to pre-inform readers.

“Abstract

Understanding public finance and budget processes has become increasingly important for the younger generation in recent years. This knowledge enables Generation Z to participate more effectively in democratic life as informed citizens. However, there is limited research on Generation Z's awareness of public budgets in the existing literature. Therefore, the study addressed a significant gap in the public finance literature. We aimed to examine the determinants influencing the state budget awareness of Generation Z undergraduates by using the survey: "Citizens' Budget Awareness and Effective Factors". Data was collected from 3,972 undergraduates across all disciplines enrolled in four Turkish state universities between December 2023 and April 2024. We conducted a multiple linear regression analysis to examine the data. The main results showed that being informed about allocated public revenues for which public services, the constitutional power of the purse, budgeting processes and budget-related transactions were the determinants that most influenced the budget awareness of the students. On the contrary, being informed about the connection between taxes and public services, the Public Financial Management and Control Law No. 5018, and the amounts of public revenues within the state budget were determined as the lowest determinants influencing their awareness. Based on our results, we mainly suggested that (i) comprehensive education programmes and interactive learning should be organised in universities by using digital tools, (ii) the link between public revenues and expenditures should be explained through awareness campaigns, and (iii) budget processes should be made more transparent and accessible.

Key words: Budget Awareness, Higher Education, Generation Z, Undergraduate Students”

5. Utilities of literature

The 78 references used in this work are all taken from journal journals. Of the 78 articles, 40 (51.28%) were published in 2018 or later (less than five years old), while 38 (48.72%) were published before 2018 (more than five years old). The paper's quality has been enhanced by the author's use of references from internationally indexed journal papers that are relevant to the topic under discussion. Seventy-five per cent of the references cited in some journals must be no more than five years old. The author might want to take this into account.

Response: Dear Reviewer, we have supported the manuscript with recent references and eliminated most of the old ones. According to the latest data, we have cited a total of 95 articles; of 77% (73) were published in 2018 and later, and 23% (22) were published before 2018.

6. Methodology

The author uses a multivariate linear regression analysis method to separate independent and dependent variables. A survey of 3,972 undergraduate students enrolled in four Turkish state universities across all subject areas between December 2023 and April 2024 was used to gather data for the analysis. While the survey data becomes the conclusions reported in the Results and Discussion section, the author may hypothetically formulate the research method employed in this section.

Response: Dear Reviewer, we have considered your suggestion in the “Results and Discussion” section. First, we have edited this sub-title as “Findings and Discussion”. Then, we have included more knowledge about multiple linear regression, as you recommended. By doing so, we emphasized the importance of this analysis technique in the research.

“We collected the data through the "Citizens' Budget Awareness and Effective Factors" survey that included a total of 13 items. While the first 12 items, as detailed in Table 2, represented as independent variables, the last item was determined the dependent variable. To analyse the data, we performed multiple linear regression analysis, a method well-suited for assessing the impact of independent variables on a dependent variable (Xue and Sun, 2019; Wu and Qiu, 2019). This analysis helps evaluate the influence of each independent variable on the dependent variable that represents the budget awareness of Generation Z. It also provided a comprehensive understanding of the relationships between the independent variables and the dependent variable. The survey was responded to by 3972 undergraduates who enrolled in various education units at four universities. Before conducting the multiple linear regression analysis, we refined our data to eliminate outlier values and reduced the risk of potential outliers by considering Cooks' distance (Cohen et al., 2013). Cooks' distance, which should be less than 1, is a diagnostic metric that evaluates how much the predicted values change when a single data point is removed, indicating that point's influence on the entire model (Khan et al., 2025).”

Additionally, we have already detailed how to formulate multiple linear regression in the sub-title “Method and Model Development.” Therefore, we have not included its formula again to avoid repetition.

7. Results and Discussions

The author of this section mixes findings and discussions. This mixing of findings and discussions makes it difficult for readers to distinguish which is the research findings data and which is the analysis results data. It is better to divide this section into two subsections, namely (sub) Results and (sub) Discussions. In the sub-results, the author can add a little about the respondent profile, followed by an analysis of the survey data, which can be presented in the form of tables and figures. Furthermore, in the sub-discussion, the author can discuss all the findings data (all tables and figures).

Response: Dear Reviewer, according to your suggestions, first we added the information about the participants’ profile.

“Among the respondents, 29.2% (1,152 undergraduates) were enrolled in the science and engineering sciences domain, 48.3% (1,905 undergraduates) in the social sciences domain, 11.8% (465 undergraduates) in the health sciences domain, 8% (316 undergraduates) in the educational sciences domain, and 2.7% (106 undergraduates) in the arts and architecture sciences domain. In addition, of the respondents, 38.4% (1,514 undergraduates) were in their first year, 27% (1,065 undergraduates) in their second year, 20.5% (809 undergraduates) in their third year, and 14.1% (556 undergraduates) in their fourth year.”

Second, we have divided “Findings and Discussions” into two sub-titles. In the “Findings” sub-title, we added the tables and figures that statistically include the research findings and have explained the statistical findings. In the “Discussion” sub-title, we debated these findings. Specifically, we have enhanced the discussion part with new references published recently. We have coloured our new contributions as colour in the text. We have not included new contributions in this letter because of being somewhat lengthy. You should follow them from the text.

8. Conclusion

The author has developed a lengthy conclusion. However, it's important to keep in mind that the conclusion serves as both a summary of the research findings and an answer to the research questions that were posed and debated during the discussion. In the Conclusion and Recommendation section, the author should include a summary of the research findings and recommendations for how interested parties might use the study's findings.

Response: Dear Reviewer, a summary of the research findings is provided in the Conclusion and Recommendations section, along with suggestions (and, by extension, the widespread impact of the study) on how interested parties can use the findings of the study. We coloured the new edits in the text as yellow.

9. Implications

The author emphasises how budgeting for Generation Z students has social and educational ramifications. Since the government is responsible for public monies, these ramifications force it to acknowledge the necessity of budgeting for Generation Z's education.

Response: Dear Reviewer, we agree with you and the study is motivated by the importance of enhancing budget awareness and literacy to increase Generation Z's ability to comprehend the social, economic, and political implications of budget policies, as mentioned in the text.

10. Quality of Communication

The paper's sections have been well-written by the author; however, keep in mind that readers like papers that are clear and cohesive.

Response: Dear Reviewer, a language expert in our institution has read the manuscript, and the text has been edited according to the English grammar style. We have not added any proof-reading certificate to protect the anonymity of the study. If necessary, we can request a certificate from our institutions. Again, we are grateful for your valuable comments and suggestions that helped us to improve this research.

---

## [Decision Letter · Decision Letter 1]

23 Mar 2025

Dear Dr. Göksu,

Thank you for submitting your manuscript to PLOS ONE. After careful consideration, we feel that it has merit but does not fully meet PLOS ONE’s publication criteria as it currently stands. Therefore, we invite you to submit a revised version of the manuscript that addresses the points raised during the review process.

2. Discussion must be enrich and add more references to support the discussion session; and

We look forward to receiving your revised manuscript.

Kind regards,

Tatchalerm Sudhipongpracha

Academic Editor

PLOS ONE

Journal Requirements:

Reviewers' comments:

Reviewer's Responses to Questions

**Comments to the Author**

Reviewer #1: All comments have been addressed

Reviewer #2: All comments have been addressed

2. Is the manuscript technically sound, and do the data support the conclusions?

Reviewer #1: Yes

Reviewer #2: Yes

3. Has the statistical analysis been performed appropriately and rigorously?

Reviewer #1: Yes

Reviewer #2: Yes

4. Have the authors made all data underlying the findings in their manuscript fully available?

Reviewer #1: Yes

Reviewer #2: Yes

5. Is the manuscript presented in an intelligible fashion and written in standard English?

Reviewer #1: Yes

Reviewer #2: Yes

Reviewer #1: The document revision 1 is completely address all comments or inputs. But one thing that must and should need more additional are;

1. Reference should be more add and need more than 8 more references and prefer from the PLOS Journal;

2. Discussion must be enrich and add more references to support the discussion session;

3. Each hypothesis should have minimum a paragraph for elaborating the idea in discussion session as a analysis.

Reviewer #2: Overall, the paper has been improved and refined by considering the suggestions and inputs from the reviewers. So in my opinion this paper is worthy of being published in the PLOS ONE Journal.

**Do you want your identity to be public for this peer review?** For information about this choice, including consent withdrawal, please see our Privacy Policy

Reviewer #1: **Yes: ** Alfrojems

Reviewer #2: **Yes: ** Buyamin

---

## [Author Response · Author response to Decision Letter 2]

9 Apr 2025

Dear Reviewers,

We are grateful for the time you have dedicated to reviewing our work and for the valuable insights you have shared. We hope that we have responded to the comments to your satisfaction. In the original version, the requested revisions are highlighted in yellow. Best regards

---

## [Decision Letter · Decision Letter 2]

27 May 2025

Dear Dr. Göksu,

Thank you for submitting your manuscript to PLOS ONE. After careful consideration, we feel that it has merit but does not fully meet PLOS ONE’s publication criteria as it currently stands. Therefore, we invite you to submit a revised version of the manuscript that addresses the points raised during the review process.

We look forward to receiving your revised manuscript.

Kind regards,

Tatchalerm Sudhipongpracha

Academic Editor

PLOS ONE

Journal Requirements:

Reviewers' comments:

Reviewer's Responses to Questions

**Comments to the Author**

Reviewer #1: All comments have been addressed

2. Is the manuscript technically sound, and do the data support the conclusions?

Reviewer #1: Yes

3. Has the statistical analysis been performed appropriately and rigorously?

Reviewer #1: Yes

4. Have the authors made all data underlying the findings in their manuscript fully available?

Reviewer #1: Yes

5. Is the manuscript presented in an intelligible fashion and written in standard English?

Reviewer #1: Yes

Reviewer #1: This article has done to address the main point purpose of the article, but this article need to add more references

**Do you want your identity to be public for this peer review?** For information about this choice, including consent withdrawal, please see our Privacy Policy

Reviewer #1: No

---

## [Author Response · Author response to Decision Letter 3]

2 Jun 2025

Dear Reviewer 1,

After your revision request, we have enhanced the manuscripts with new references. Now, the reference number has reached a total of 118 in its last version. By doing so, the paper has significantly improved in the introduction and especially the literature review parts. We have colored the new contributions yellow. We thank you for reviewing our manuscript again.

---

## [Decision Letter · Decision Letter 3]

6 Jul 2025

Decoding Budget Awareness: A Multivariate Analysis of Generation Z Undergraduates

PONE-D-25-02233R3

Dear Dr. Göksu,

We’re pleased to inform you that your manuscript has been judged scientifically suitable for publication and will be formally accepted for publication once it meets all outstanding technical requirements.

Kind regards,

Tatchalerm Sudhipongpracha

Academic Editor

PLOS ONE

Additional Editor Comments (optional):

Reviewers' comments:

Reviewer's Responses to Questions

**Comments to the Author**

Reviewer #1: All comments have been addressed

2. Is the manuscript technically sound, and do the data support the conclusions?

Reviewer #1: Yes

3. Has the statistical analysis been performed appropriately and rigorously?

Reviewer #1: Yes

4. Have the authors made all data underlying the findings in their manuscript fully available?

Reviewer #1: Yes

5. Is the manuscript presented in an intelligible fashion and written in standard English?

Reviewer #1: Yes

Reviewer #1: All issued are complete done and the article is proper for publi, just need a minor revised particularly for the references of the PLOS journal

**Do you want your identity to be public for this peer review?** For information about this choice, including consent withdrawal, please see our Privacy Policy

Reviewer #1: No

---

## [Editor Report · Acceptance letter]

PONE-D-25-02233R3

PLOS ONE

Dear Dr. Göksu,

I'm pleased to inform you that your manuscript has been deemed suitable for publication in PLOS ONE. Congratulations! Your manuscript is now being handed over to our production team.

Kind regards,

on behalf of

Dr. Tatchalerm Sudhipongpracha

Academic Editor

PLOS ONE